DATA RELEASE

# Spatial temporal distribution of *Anopheles* mosquitoes in different ecological zones of Ghana

Anisa Abdulai[1], Christopher Mfum Owusu-Asenso[1],
Abdul Rahim Mohammed Sabtiu[1], Isaac Kwame Sraku[1],
Yaw Akuamoah-Boateng[1], Abena Ahema Ebuako[1], Lourees Esi Awotwe[1],
Richard Tettey Doe[1], Emmanuel Nana Boadu[1], Akua Aboagyewaa Appiah[1],
Grace Arhin Danquah[1], Dhikrullahi Bunkunmi Shittu[1],
Gabriel Akosah-Brempong[2], Cosmos Manwovor-Anbon Pambit Zong[2],
Daniel Kodjo Halou[3], Osei Kwaku Akuoko[4], Akua Obeng Forson[5] and
Yaw Asare Afrane[1,*]

1   Centre for Vector-Borne Disease Research, Department of Medical Microbiology, University of Ghana Medical School, Korle-Bu, Accra, Ghana
2   African Regional Postgraduate Programme in Insect Sciences (ARPPIS), Department of Animal Biology and Conservation Science, College of Basic and Applied Sciences, University of Ghana, Accra, Ghana
3   Department of Vector Biology, Liverpool School of Tropical Medicine, UK
4   Department of Parasitology, Noguchi Memorial Institute for Medical Research, College of Health Sciences, University of Ghana, Legon, Accra, Ghana
5   Department of Medical Laboratory Science, School of Biomedical and Allied Health Sciences, University of Ghana, Korle-Bu, Accra, Ghana

**Submitted:** 23 September 2025

\* Corresponding author. E-mail: yafrane@ug.edu.gh

Preprint submitted at https://doi.org/10.60763/africarxiv/10161

Included in the series: ***Vectors of human disease*** (https://doi.org/10.46471/GIGABYTE_SERIES_0002)

## ABSTRACT

Vector control is a cornerstone for malaria management in Sub-Saharan Africa. Understanding the distribution dynamics and ecology of major malaria vectors is important for strengthening the current control efforts of national malaria control programmes. This project monitored the spatiotemporal distribution of *Anopheles* mosquitoes across different ecological zones of Ghana from 2017 to 2025. *Anopheles* mosquitoes were sampled from twelve sites across the three ecological zones of Ghana (Coastal, Forest and Sahel Savannah zones) using human landing catches and Prokopack aspirators. Mosquitoes were subjected to morphological and molecular species identification. Sporozoite infection rates and blood meal sources of collected blood fed female mosquitoes were both assessed using PCR. A total of 47,771 Anopheline mosquitoes (*An. gambiae* s.l, *An. funestus*, *An. pharoensis* and *An. rufipes*) were collected across the three ecological zones. *Anopheles gambiae* s.l, and particularly *An. coluzzii* and *An. gambiae* s.s were the predominant species across the study sites and ecological zones. Sporozoite infections were higher in the forest and sahel zones compared to the coastal zone, and the overall human blood index was 40.46%. Our findings provide relevant data for improving current vector control for malaria in Ghana.

**Subjects** Ecology, Biodiversity, Taxonomy

# DATA DESCRIPTION
## Background and context

Malaria remains a major public health issue in Sub-Saharan Africa (SSA), accounting for over 251 million cases and over 500,000 deaths [1]. Over 94% of all malaria cases are

reported in the WHO Africa Region [2]. Ghana is one of the 11 African countries contributing approximately two-thirds of all global malaria cases [3]. Despite other malaria control efforts, vector control remains the mainstay for malaria control and elimination in Ghana and across SSA. Over the last two decades, vector control tools like long-lasting insecticide treated nets (LLINs) and indoor residual spraying (IRS) have played a key role in the reduction of malaria transmission in SSA [4, 5]. However, increasing insecticide resistance in malaria vectors threatens the effectiveness of current control efforts [6–8].

There is a need for new vector interventions for malaria control and elimination. New frontiers in vector control are emerging with innovative tools in the pipeline such as attractive sugar baits, spatial repellents, endectocides and gene drive [9–11]. However, the success of current and emerging innovations in vector control has largely hinged on our deep understanding of the malaria vector distribution and ecology [12, 13]. In Ghana, *Anopheles gambiae* complex and *Anopheles funestus* complex are the main malaria vectors [14–17]. These complexes all have a diverse distribution, behaviour and ecology [15, 17, 18], and furthermore ecological shifts in vector composition and shifts in vector behaviour have been reported [19, 20]. This suggests that the distribution and ecology of local malaria vectors may vary from one geographical area to another due to different environmental factors [13, 21].

Understanding the distribution and ecology of local malaria vectors will therefore be useful in the development of targeted vector interventions for malaria control in Ghana. This study details the spatiotemporal distribution of *Anopheles gambiae* s.l across different ecological zones of Ghana. These ecological zones differing in climatic and environmental conditions which may affect the ecology and distribution of local malaria vectors.

## METHODS

### Study sites

This study was conducted in twelve sites across the three main ecological zones of Ghana (Coastal, Forest and Sahel Savannah zone) during the dry season (February–March) and the rainy season (May–July) from 2017 to 2025. These sites are Anyakpor, Dodowa, Aflao, Takoradi, Elubo, Tuanikorpe, Pediatorkope, Allorkpem, Dwease, Kpalsogu, Pagaza and Libga (Figure 1).

Anyakpor (5° 46′ 51.96″ N, 0° 35′ 12.84″ E), Tuanikorpe (05°50.029′ N, 000°38.778′ E), Pediatorkope (5° 48′ 52.6″ N 0° 37′ 66.00″ E), Allorkpem (05°47.960N, 000°39.180W) are located in the coastal zone of Ghana. These sites are characterized by dry climates with temperatures within 23 °C to 28 °C and a bimodal rainfall pattern with a long rainy season and a short rainy season [22]. Anyakpor is a farming community located in the Ada East District. Irrigated farming is practiced all year round in that site, leading to the creation of numerous mosquito breeding sites, and hence high densities of mosquitoes [15]. Tuanikope, Pediatorkope, and Allorkpem are island communities along the Volta Lake in the Dagme East district of Greater Accra where the main occupation is fishing. These sites have been difficult to access, hence they have not received vector interventions such as IRS and LLINs for several years [23].

Dwease (06°32.383N, 001°14.368W) is a village located in the Asante-Akim Central municipality and characterized by a wet-semi equatorial climate and a bimodal rainfall pattern. Dwease has a semi-deciduous forest vegetation and their main occupation is farming. Kpalsogu (09°34.019N, 001°01.975W), Pagaza (9° 22′ 33″ N, 0° 42′ 30″ W) and Libga



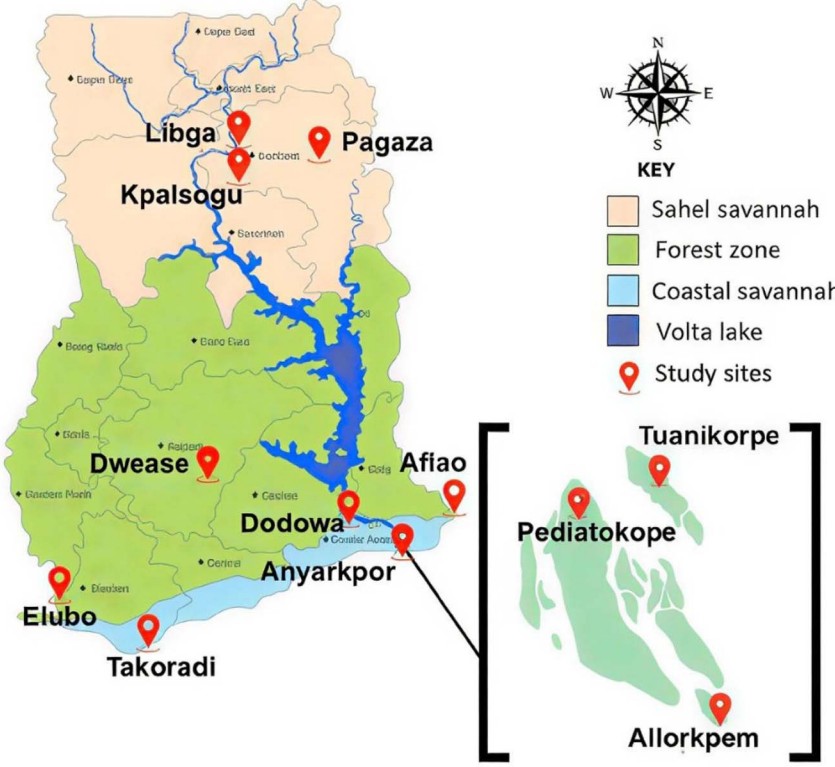

**Figure 1.** Map of study sites across the three ecological zones of Ghana.

(09°35.567N, 000°50.743W) are rural communities in the Kumbungu and Tamale municipalities in the Sahel savannah zone of Ghana. These sites have annual temperatures, ranging from 28 °C to 42 °C and a unimodal rainfall pattern (from May to November). Irrigated farming is practiced throughout the year creating many swamps which are suitable breeding habitats for Anopheles mosquitoes. Malaria interventions such as SMCs and IRS are carried out in Kpalsogu [5], and insecticide-treated nets are distributed in Libga and Pagaza.

## Mosquito collections

Adult mosquitoes were collected using human landing catches (HLCs) and Prokopack aspirators and CDC light traps. Anopheles mosquitoes were collected using HLC from 18:00 to 06:00 hours for four consecutive days in 16 randomly selected houses per site and season using well -described protocols [18, 22]. CDC light traps were also set up in indoor and outdoor of houses to collect adult mosquitoes from 18:00 to 06:00. For fair representation, each site was split into four sections and one-day sampling was conducted in each section. Collectors were rotated hourly as a quality control measure to reduce collection bias. Furthermore, mechanical Prokopack aspirators were used to collect adult mosquitoes from up to 20 houses (at least 50 m apart from each other) from each study site from 06:00 and 08:00. Indoor and outdoor collectors were done simultaneously in each house. Collected mosquitoes were stored in eppendorf tubes with silica gel, placed in a cooler box containing ice and transported to the insectary.

**Table 1.** Primer sequences used for PCR analysis.

| Primer name | Sequence (5'-3') | Reference |
| --- | --- | --- |
| **Species identification** | | |
| UN | GTGTGCCGCTTCCTCGATGT | |
| AG | CTGGTTTGGTCGGCACGTTT | |
| AR | AAGTGTCCTTCTCCATCCTA | |
| AM | GTGACCAACCCACTCCCTTGA | |
| F6.1a | TCGCCTTAGACCTTGCGTTA | |
| R6.1b | CGCTTCAAGAATTCGAGATAC | [26] |
| **Sporozoites** | | |
| COX-IF | AGAACGAACGCTTTTAACGCCTG | |
| COX-IR | ACTTAATGGTGGATATAAAGTCCATCCwGT | [27] |
| **Blood meals** | | |
| Goat | CCTAATCTTAGTACTTGTACCCTTCCTC | |
| Human | GGCTTACTTCTCTTCATTCTCTCCT | |
| Pig | CCTCGCAGCCGTACATCTC | |
| Dog | GGAATTGTACTATTATTCGCAACCAT | |
| Cow | CATCGGCACAAATTTAGTCG | |
| UNFOR403 | TGAGGACAAATATCATTCTGAGG | |
| UNREV1025 | GGTTGTCCTCCAATTCATGTTA | [28] |

## Morphological and molecular species identification of Anopheles mosquitoes

Collected mosquitoes were morphologically identified using morphological keys by Gillies and Coetzee [24]. DNA was extracted from the legs of mosquitoes using the QIAGEN DNeasy Blood and Tissue Kits according to the manufacturer's instructions. Members of the *Anopheles gambiae* s.l complex were further distinguished molecularly using PCR. Species-specific primers for *Anopheles gambiae, An. arabiensis, An. melas* and a universal primer were used for the PCR reactions following protocols by Scott *et al.* [25]. *Anopheles gambiae* s.s *and An. coluzzii* were further distinguished by PCR-RFLP using the method by Fanello *et al.* [26].

## Sporozoites infectivity of adult *Anopheles gambiae* s.l mosquitoes

DNA was extracted from the head and thorax of adult *An. gambiae* s.l mosquitoes were used for the detection of *Plasmodium falciparum* sporozoites using PCR using well-described protocols by Echeverry *et al.* [27]. Species-specific primers targeting the cytochrome oxidase I (COX-I) of *P. falciparum* sporozoites were used for the PCR assays.

## Blood meal analysis of blood fed resting Anopheles mosquitoes

DNA was extracted from the abdomens of bloodfed *An. gambiae* s.l mosquitoes using QIAGEN DNeasy Blood and Tissue Kits according to the manufacturer's instructions. The blood meals of engorged female mosquitoes were determined using PCR with well-described protocols from Kent and Norris [28]. Host-specific primers (human, cow, goat, pig and dog) were used for the PCR amplifications (Table 1). Positive controls were included for each host in the PCR analyses, and laboratory-reared unfed *An. gambiae* were used as the negative control.

## Data analysis

Chi-square tests/fisher exact tests were used to assess differences in seasonal abundance and species composition of malaria vectors collected by location, study site and season.

**Table 2.** Abundance and seasonal distribution of Anopheline vectors across the study sites.

| Ecozone | Sites | Anophelines species | | | | |
|---|---|---|---|---|---|---|
| | | *An. gambiae* s.l | *An. funestus* | *An. pharoensis* | *An. rufipes* | **Total** |
| **Dry season** | | | | | | |
| Coastal | Anyakpor | 3247 | 0 | 46 | 3 | 3296 |
| | Dodowa | 832 | 0 | 0 | 1 | 833 |
| | Aflao | 10 | 0 | 0 | 0 | 10 |
| | Elubo | 0 | 0 | 0 | 0 | 0 |
| | Tuanikorpe | 446 | 0 | 45 | 39 | 530 |
| | Pediatorkope | 162 | 0 | 0 | 72 | 234 |
| | Allorkpem | 271 | 0 | 2 | 40 | 313 |
| Forest | Dwease | 1443 | 0 | 1 | 0 | 1444 |
| Sahel zone | Kpalsogu | 8583 | 1 | 182 | 170 | 8936 |
| | Pagaza | 532 | 0 | 0 | 0 | 532 |
| | Libga | 32 | 0 | 0 | 0 | 32 |
| | **Total** | **15,558** | **1** | **276** | **325** | **16,160** |
| **Wet season** | | | | | | |
| Coastal | Anyakpor | 2055 | 0 | 491 | 2 | 2548 |
| | Dodowa | 6985 | 1 | 19 | 2 | 7007 |
| | Aflao | 44 | 0 | 0 | 0 | 44 |
| | Takoradi | 35 | 0 | 0 | 0 | 35 |
| | Elubo | 37 | 0 | 0 | 0 | 37 |
| | Tuanikorpe | 128 | 0 | 11 | 6 | 145 |
| | Pediatorkope | 71 | 0 | 7 | 4 | 82 |
| | Allorkpem | 145 | 0 | 0 | 10 | 155 |
| Forest | Dwease | 2262 | 62 | 1 | 0 | 2325 |
| | Abetifi | 17 | 0 | 0 | 0 | 17 |
| Sahel zone | Kpalsogu | 8174 | 80 | 769 | 21 | 9044 |
| | Pagaza | 7653 | 572 | 36 | 3 | 8264 |
| | Libga | 1904 | 4 | 0 | 0 | 1908 |
| | **Total** | **29,510** | **719** | **1334** | **48** | **31,611** |
| | **Grand total** | **45,068 (94.34%)** | **720 (1.51%)** | **1610 (3.37%)** | **373 (0.78%)** | **47,771 (100)** |

Sporozoite infection rates were calculated as the proportion of mosquitoes testing positive for Plasmodium sporozoites, obtained by dividing the number of sporozoite-positive mosquitoes by the total number examined. Human blood index (HBI) was calculated as the proportion of blood-fed mosquitoes that had fed on humans relative to the total number analyzed for blood meal origin.

## RESULTS

## Abundance and seasonal distribution of Anopheline vectors

Overall, a total of 47,771 Anopheline mosquitoes (*An. gambiae* s.l, *An. funestus*, *An. pharoensis* and *An. rufipes*) were collected from all the study during the dry and rainy seasons. More mosquitoes were collected in the rainy (31,611/47,771, 66.17%) season compared to the dry season (16,160/47,771, 33.83%). *Anopheles gambiae* s.l was predominant across both rainy and dry seasons (45,068/47,771, 94.34%) followed by *An. pharoensis* (1610/47,771, 3.37%), *An. funestus* (720/47,771, 1.51%) and *An. rufipes* (373/47,771, 0.78%) (Table 2).

## Indoor and outdoor distribution of *An. gambiae* s.l vectors

More *An. gambiae* s.l were collected in the rainy season (65.5%, 29,510/45,068) compared to the dry season (34.5%, 15,558/45,068). *Anopheles gambiae* s.l mosquitoes were predominantly collected outdoors in both the dry season (61.3%, 9,540/15,558) and rainy season (55.8%, 16,476/29,510). In Anyakpor (from the coastal zone), more *An. gambiae* s.l mosquitoes were collected indoors in both the dry season (58.8%, 1909/3247) and the rainy season (56.2%, 1154/2055). However, in Kpalsogu (Sahel zone), more *An. gambiae* s.l

**Table 3.** Indoor and outdoor distribution of *An. gambiae* s.l across the study sites.

| Eco-zones | Sites | Dry season *N* (%) | | | Wet season *N* (%) | | |
|---|---|---|---|---|---|---|---|
| | | Total | Indoor | Outdoor | Total | Indoor | Outdoor |
| Coastal zone | Anyakpor | 3247 (100) | 1909 (58.8) | 1338 (41.2) | 2055 (100) | 1154 (56.2) | 901 (43.8) |
| | Dodowa | 832 (100) | 392 (47.1) | 440 (52.9) | 6985 (100) | 3125 (44.7) | 3860 (55.3) |
| | Elubo | 0 | 0 | 0 | 37 (100) | 19 (51.4) | 18 (48.6) |
| | Aflao | 10 (100) | 0 | 10 (100) | 44 (100) | 12 (27.3) | 32 (72.7) |
| | Takoradi | 0 | 0 | 0 | 35 (100) | 8 (22.9) | 27 (77.1) |
| | Tuanikorpe | 446 (100) | 223 (50) | 223 (50) | 128 (100) | 64 (50) | 64 (50) |
| | Pediatorkope | 162 (100) | 112 (69.1) | 50 (30.9) | 71 (100) | 55 (77.5) | 16 (22.5) |
| | Allorkpem | 271 (100) | 178 (65.7) | 93 (34.3) | 145 (100) | 88 (60.7) | 57 (39.3) |
| Forest | Dwease | 1443 (100) | 748 (51.8) | 695 (48.2) | 2262 (100) | 1244 (55) | 1018 (45) |
| | Abetifi | 0 | 0 | 0 | 17 (100) | 0 | 17 (100) |
| Sahel zone | Kpalsogu | 8583 (100) | 2236 (26.1) | 6347 (73.9) | 8174 (100) | 3049 (37.3) | 5125 (62.7) |
| | Pagaza | 532 (100) | 210 (39.5) | 322 (60.5) | 7653 (100) | 3812 (49.8) | 3841 (50.2) |
| | Libga | 32 (100) | 10 (31.3) | 22 (68.8) | 1904 (100) | 404 (21.2) | 1500 (78.8) |
| | **Overall total** | **15,558 (100)** | **6018 (38.7)** | **9540 (61.3)** | **29,510 (100)** | **13,034 (44.2)** | **16,476 (55.8)** |

mosquitoes were collected outdoors in both the dry season (73.9%, 6347/8583) and rainy seasons (62.7%, 5125/8174) (Table 3).

## Species composition of *Anopheles gambiae* s.l mosquitoes across the study sites

A subset of 2,730 *An. gambiae* s.l were morphologically identified, out of which *An. coluzzii* (53.33%, 1456/2730) was the predominant species, followed by *Anopheles gambiae* s.s (43.7%, 1193/2730), *An. arabiensis* (1.45%, 39/2730), *An. melas* (1.35%, 37/2730) and *An. gambiae/An. coluzzii* hybrids (0.18%, 5/2730). In the coastal sites, *An. coluzzii* was the predominant sites in Anyakpor (83.5%) and Pediatorkope (77.39%). However, *An. gambiae* s.s was the predominant species in Dodowa (87.02%), Tuanikorpe (63.74%) and Allorkpem (78.95%). For the forest and sahel zone sites, *An. coluzzii* was the most common species (>80%) in all the sites except in Pagaza, where *An. gambiae* s.s was most predominant (72.43%) (Figure 2). Significant differences in the distribution of *An. gambiae* s.l was observed across the study sites ($\chi^2$ = 301.06, *df* = 8, *P* < 0.001).

## Indoor and outdoor distribution of *An. gambiae* s.l from different ecological zones

In the coastal zone, a the greatest number of indoor collected mosquitoes were *An. coluzzii* (52.8%, 480/909) while a majority of outdoor collected mosquitoes were identified as *An. gambiae* s.s (52.4%, 393/750). For the forest and Sahel savannah zones, *An. coluzzii* was the predominant species collected both indoors and outdoors (*P* < 0.05) (Table 4). *Anopheles melas* was detected only in the coastal zone (0.3%) and *An. arabiensis* were more predominant outdoors (6.3%) compared to indoors (3.3%) and was only detected in the sahel zone (Table 4).

## Sporozoites infectivity rates of *An. gambiae* s.l mosquitoes

A total of 1,062 *An. gambiae* s.l. mosquitoes were tested for the presence of *P. falciparum* circumsporozoite protein (CSP), of which 3.11% (33/1062) tested positive. Significantly higher sporozoite rates were detected in *An. gambiae* s.l collected during the wet season (4.46%, 19/426) compared to the dry season (2.2%, 14/636) (Fishers exact, *P* < 0.05). Sporozoite infection rates were higher in indoor mosquitoes (3.45%, 16/448) compared to

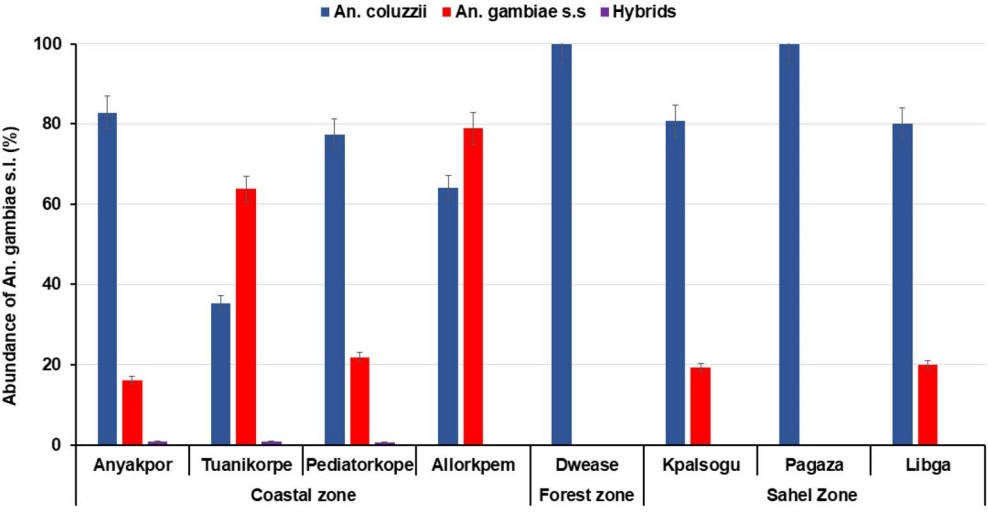

**Figure 2.** Species discrimination of *Anopheles gambiae* s.l across the study sites.

**Table 4.** Indoor and outdoor distribution of *An. gambiae* s.l. across the ecological zones of Ghana.

| Ecozones | Location | | *An. coluzzii* | *An. gambiae* s.s | *An. melas* | *An. arabiensis* | Hybrids |
|---|---|---|---|---|---|---|---|
| | | | | *An. gambiae* s.l *N* (%) | | | |
| Coastal | Indoor | 909 (100) | 480 (52.8) | 412 (45.3) | 12 (1.3) | 0 | 5 (0.6) |
| | Outdoor | 750 (100) | 332 (44.3) | 393 (52.4) | 25 (3.3) | 0 | 0 |
| | **Total** | **1659 (100)** | **812 (49)** | **805 (48.5)** | **37 (2.2)** | **0** | **5 (0.3)** |
| Forest | Indoor | 131 (100) | 115 (87.8) | 16 (12.2) | 0 | 0 | 0 |
| | Outdoor | 134 (100) | 103 (77) | 31 (23) | 0 | 0 | 0 |
| | **Total** | **265 (100)** | **218 (82.3)** | **47 (17.7)** | **0** | **0** | **0** |
| Sahel | Indoor | 395 (100) | 199 (50.4) | 183 (46.3) | 0 | 13 (3.3) | 0 |
| | Outdoor | 411 (100) | 218 (53.1) | 167 (40.6) | 0 | 26 (6.3) | 0 |
| | **Total** | **806 (100)** | **417 (51.8)** | **350 (43.42)** | **0** | **39 (4.8)** | **0** |

*N* = total number of samples tested, % = percentage, Hybrids = *An. coluzzii/An. gambiae* s.s.

outdoor *An. gambiae* s.l. mosquitoes (2.84%, 17/598) (*P* > 0.05) (Table 5). Higher sporozoite infection rates were detected in *Anopheles gambiae* s.l from the forest (7.69%, 1/13) and Sahel Savannah zone (4.11%, 6/146) compared to the coastal zone (2.88%, 26/903) (*P* > 0.05). Significantly higher sporozoite rates were detected in *An. coluzzii* (4.6%, 24/522) compared to *An. gambiae* s.s. mosquitoes (1.68%, 9/535) (Fishers exact, *P* < 0.05). The highest sporozoite infection rates in indoor collected mosquitoes were detected in Dwease (12.5%) and Libga (5%). However, for outdoor collected mosquitoes, sporozoite infections were only observed in Tuanikorpe (5.33%), Allorkpem (2.44%), Kpalsogu (4.55%) and Libga (4%) (Table 5).

## Blood meal sources of *An. gambiae* s.l mosquitoes

A total of 173 blood-fed female *An. gambiae* s.l. mosquitoes were tested for their blood meal sources. Overall, the Animal Blood Index (ABI) was higher (41.04%, 71/173) than Human Blood Index (HBI) (40.46%, 70/173). In the coastal zone, all blood fed mosquitoes were *An. coluzzii*, with a higher HBI in indoor (63.64%) compared to outdoor *An. coluzzii* mosquitoes (50%). In the forest zone, all indoor and 60% of outdoor blood-fed *An. coluzzii* tested had human blood meals. In the Sahel zone, a higher HBI and ABI was observed in

**Table 5.** Sporozoite infection rates detected in *An. gambiae* s.l. from the study sites.

| Study site | Indoor | | | Outdoor | | |
|---|---|---|---|---|---|---|
|  | *N* | Pf CSP positives | SR (%) | *N* | Pf CSP positives | SR (%) |
| Anyakpor | 61 | 0 | 0 | 50 | 0 | 0.00 |
| Tuanikorpe | 173 | 6 | 3.47 | 169 | 9 | 5.33 |
| Pediatorkope | 127 | 4 | 3.15 | 19 | 0 | 0.00 |
| Allorkpem | 181 | 4 | 2.21 | 123 | 3 | 2.44 |
| Dwease | 8 | 1 | 12.5 | 5 | 0 | 0.00 |
| Kpalsogu | 27 | 1 | 3.70 | 66 | 3 | 4.55 |
| Pagaza | 1 | 0 | 0.00 | 7 | 0 | 0.00 |
| Libga | 20 | 1 | 5.00 | 25 | 1 | 4.00 |
| **Total** | **598** | **17** | **2.84** | **464** | **16** | **3.45** |

SR = Sporozoite rate, *N* = total number of samples tested, % = percentage, Pf CSP = *P. falciparum* circumsporozoite protein (CSP).

**Table 6.** Blood meal sources of blood fed *An. gambiae* s.l across different ecological zones in Ghana.

| Ecological zones | Blood-meal origins | *An. coluzzii N* (%) | | *An. gambiae s.s N* (%) | |
|---|---|---|---|---|---|
|  |  | Indoor | Outdoor (%) | Indoor (%) | Outdoor (%) |
| Coastal zone | Number tested | 11 (100) | 4 (100) | 0 | 0 |
|  | Human | 5 (45.45) | 1 (25) | 0 | 0 |
|  | Goat | 3 (27.27) | 2 (50) | 0 | 0 |
|  | Human + Goat | 1 (9.09) | 0 | 0 | 0 |
|  | Human + Pig | 1 (9.09) | 1 (25) | 0 | 0 |
|  | Unidentified | 1 (9.09) | 0 | 0 | 0 |
|  | HBI | 63.64 | 50 | 0 | 0 |
|  | ABI | 45.45 | 75 | 0 | 0 |
| Forest zone | Number tested | 8 (100) | 5 (100) | 0 | 0 |
|  | Human | 8 (100) | 3 (60) | 0 | 0 |
|  | Unidentified | 0 | 2 (40) | 0 | 0 |
|  | HBI | 100 | 60 | 0 | 0 |
|  | ABI | 0 | 0 | 0 | 0 |
| Sahel zone | Number tested | 36 (100) | 83 (100) | 12 (100) | 14 (100) |
|  | Human | 17 (47.22) | 20 (24.10) | 6 (50) | 5 (35.71) |
|  | Cow | 8 (22.22) | 25 (30.12) | 2 (16.67) | 3 (21.43) |
|  | Goat | 4 (11.22) | 5 (6.02) | 2 (16.67) | 3 (21.43) |
|  | Dog | 0 | 2 (2.41) | 0 | 1 (7.14) |
|  | Human + Cow | 0 | 2 (2.41) | 0 | 0 |
|  | Dog + Goat | 1 (2.77) | 0 | 0 | 0 |
|  | Cow + Goat | 4 (11.22) | 1 (1.20) | 0 | 0 |
|  | Unidentified | 2 (5.55) | 28 (33.73) | 2 (16.67) | 2 (14.29) |
|  | HBI | 47.22 | 26.51 | 50.00 | 35.71 |
|  | ABI | 47.22 | 42.17 | 33.33 | 50.00 |

*N* = total number of samples tested, % = percentage, HBI = Human Blood Index, ABI = Animal Blood Index.

indoor collected *An. coluzzii* mosquitoes (HBI and ABI = 47.22) compared to outdoor collected mosquitoes (HBI = 26.51, ABI = 42.17). This was similar for indoor blood-fed *An. gambiae* s.s. mosquitoes from the Sahel zone, however ABI was higher outdoors (50%) compared to indoors (33.33) (Table 6).

## RE-USE POTENTIAL

Understanding the distribution of the major malaria vector, *An. gambiae* s.l. is of key relevance for malaria control and elimination [12]. Our study identified four Anophelines species (*An. gambiae* s.l, *An. funestus*, *An. pharoensis* and *An. rufipes)* at varying abundance across different sites in the three major ecological zones of Ghana. *Anopheles gambiae* s.l, particularly *An. coluzzii* and *An. gambiae* s.s were predominant across the study sites. These species are very predominant across Ghana and are known to have very wide geographical spread and diverse habitats [15–18]. Other species such as *Anopheles melas* and *Anopheles*



*arabiensis* are localised to particular parts of Ghana [15, 29]. *Anopheles melas* is mostly found in the coastal areas because it breeds in brackish water while *An. arabiensis* is found exclusively in the Sahel zone of Ghana due to the drier arid environment [15, 18, 22]. The dynamics of vector distribution varied in *An. coluzzii* and *An. gambiae* s.s in the Coastal zone with a higher abundance of *An. gambiae* s.s found only in the dry season compared to the rainy season.

Sporozoite infection rates were higher in the wet season compared to the dry season, in line with several studies across Ghana and Africa [30, 31]. Sporozoite infection rates were high in the Sahel and Forest zones compared to the coastal zone. Malaria prevalence was high in the Sahel Savannah zone, with a seasonal pattern peaking during the rainy season from July to November [32]. HBI was higher indoors compared to outdoors mosquitoes in *An. coluzzii*, which is in line with another study in Ghana that reported higher HBI in indoor compared to outdoor mosquitoes [18]. The observation of *An. gambiae* s.s in the sahel zone, where more outdoor mosquitoes fed on humans, may suggest potential shifts in biting behaviour in *Anopheles gambiae* mosquitoes, as reported in other studies [19, 20, 33]. These shifts may be due to changes in human behaviour or the high coverage of indoor based chemical vector interventions like LLINs and IRS [20, 34, 35].

This data provides key insights on *An. gambiae* s.l distribution, sporozoite infection rates and blood meal sources across Ghana and can be used for optimize current vector control strategies implemented by the National Malaria Elimination Programme to adapt targeted approaches for malaria control.

## DATA VALIDATION AND QUALITY CONTROL

All mosquito collections were supervised by well experienced entomologists. Measures such as rotations and quadrant sampling of each site was done to avoid bias. All mosquitoes were morphologically and molecularly identified by well experienced personnel using standard techniques. The final dataset was validated in the Integrated Publishing Toolkit (IPT) of the Global Biodiversity Information Facility (GBIF) [36]. The IPT validated the data through its network and the metadata can be found in GBIF [37].

## DATA AVAILABILITY

All supporting data for this article are published through the Integrated Publishing Toolkit of GBIF and are available under a CC0 waiver from GBIF [37].

## EDITOR'S NOTE

This paper is part of a series of Data Release articles working with GBIF and supported by TDR, the Special Program for Research and Training in Tropical Diseases, hosted at the World Health Organization [38].

## ABBREVIATIONS

ABI: Animal Blood Index; CDC: Center for Disease Control; CSP: Circumsporozoite Protein; GBIF: Global Biodiversity Information Facility; HLC: Human Landing Catch; HBI: Human Blood Index; IPT: Integrated Publishing Toolkit; IRS: Indoor Residual Spraying; LLINs: Long-Lasting Insecticide Treated Nets; PCR: Polymerase Chain Reaction; SMCs: Seasonal Malaria Chemoprevention; SSA: Sub-Saharan Africa.

## DECLARATIONS

### Ethics approval and consent to participate

Ethical Approval was obtained from the Ghana Health Service ethics and Review Committee (GHS-ERC number: GHS-ERC: 051/03/23). Prior to the start of the study, meetings were held at each site with chiefs, community leaders, and residents to introduce the project. Approval to conduct the research within the communities was obtained from the respective leaders. Written informed consent was secured from all adult volunteers participating in the HLC study before training and mosquito collection activities. In addition, verbal consent was obtained from household heads for mosquito sampling within their homes and compounds.

### Competing interests

The authors declare no competing interests.

### Authors' contributions

AAE, LEA, AA, AOF and YAA were responsible for the study design, supervised the data collection, and contributed to the writing of the manuscript. AA, AAE, LEA, CMO-A, ARMS, IKS, RTD, ENB, GAD, SBS, GA-B, CM-APZ, DKH, OKA and YAB performed the data collection and laboratory work. AA, CMO-A, AAE, LEA contributed to the analysis of the data. AA drafted the manuscript. All the authors read and approved the final manuscript.

### Funding

This study was supported by grants from the National Institute of Health (https://www.nih.gov/) received by YAA (Grant numbers: R01 A1123074 and D43 TW 011513).

### Acknowledgements

Special thanks to the residents and community leaders of our study sites for giving us access to sample mosquitoes in their communities. We are also grateful to all the community field assistants for helping in the sampling and the entire team of the Centre for Vector-Borne Disease Research, Department of Medical Microbiology, University of Ghana Medical School.

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
