## [Editor Report]

Editor’s AssessmentIn Sub Saharan Africa malaria is a major public health challenge accounting for over 251 million cases and over 500,000 deaths, there is however limited data on spatiotemporal distribution of Anopheles mosquitoes across different ecological zones, which hinders modelling research and control strategies. This paper is one of a series of Data Release papers in GigaByte supported by TDR and the WHO describing datasets hosted in GBIF to tackle these data gaps in vectors of human disease data. This paper presents data on Anopheles mosquitoes sampled from twelve sites across the three ecological zones of Ghana (Coastal, Forest and Sahel Savannah zones) using human landing catches and Prokopack3 aspirator. Overall, a total of 47,771 Anopheline mosquitoes (An. gambiae s.l, An. funestus, An. Pharoensis and An. rufipes were collected from all the study during the dry and rainy seasons. Peer review and data auditing found the data to be well validated. The information contained can serve as a resource for studies focused on assessing transmission risks, vector control strategies, disease surveillance and a broader comprehension of mosquito ecology in the various ecological zones of Ghana.Editor’s AssessmentIn Sub Saharan Africa malaria is a major public health challenge accounting for over 251 million cases and over 500,000 deaths, there is however limited data on spatiotemporal distribution of Anopheles mosquitoes across different ecological zones, which hinders modelling research and control strategies. This paper is one of a series of Data Release papers in GigaByte supported by TDR and the WHO describing datasets hosted in GBIF to tackle these data gaps in vectors of human disease data. This paper presents data on Anopheles mosquitoes sampled from twelve sites across the three ecological zones of Ghana (Coastal, Forest and Sahel Savannah zones) using human landing catches and Prokopack3 aspirator. Overall, a total of 47,771 Anopheline mosquitoes (An. gambiae s.l, An. funestus, An. Pharoensis and An. rufipes were collected from all the study during the dry and rainy seasons. Peer review and data auditing found the data to be well validated. The information contained can serve as a resource for studies focused on assessing transmission risks, vector control strategies, disease surveillance and a broader comprehension of mosquito ecology in the various ecological zones of Ghana.

---

## [Reviewer Report]

Upload additional filesDRR-202509-06-R01/stage_files/DRR-202509-06/Review MS/GBIF-Data-Review-DRR-202509-06.pdfReviewer name and names of any other individual's who aided in reviewer Yannan FanDo you understand and agree to our policy of having open and named reviews, and having your review included with the published papers. (If no, please inform the editor that you cannot review this manuscript.)YesIs the language of sufficient quality?YesPlease add additional comments on language quality to clarify if needed
Are all data available and do they match the descriptions in the paper? NoAdditional CommentsThe Occurence in the GBIF dataset is 1062, but the MS described as 47771. Please update the records in the GBIFAre the data and metadata consistent with relevant minimum information or reporting standards? See GigaDB checklists for examples <a href="http://gigadb.org/site/guide" target="_blank">http://gigadb.org/site/guide</a>YesAdditional CommentsIs the data acquisition clear, complete and methodologically sound?YesAdditional CommentsIs there sufficient detail in the methods and data-processing steps to allow reproduction?YesAdditional CommentsIs there sufficient data validation and statistical analyses of data quality? Not my area of expertiseAdditional CommentsIs the validation suitable for this type of data?YesAdditional CommentsIs there sufficient information for others to reuse this dataset or integrate it with other data?YesAdditional CommentsAny Additional Overall Comments to the AuthorRecommendationMinor Revision

---

## [Reviewer Report]

Reviewer name and names of any other individual's who aided in reviewer Idelphonse Bonaventure AhogniDo you understand and agree to our policy of having open and named reviews, and having your review included with the published papers. (If no, please inform the editor that you cannot review this manuscript.)YesIs the language of sufficient quality?YesPlease add additional comments on language quality to clarify if needed
Are all data available and do they match the descriptions in the paper? YesAdditional CommentsAre the data and metadata consistent with relevant minimum information or reporting standards? See GigaDB checklists for examples <a href="http://gigadb.org/site/guide" target="_blank">http://gigadb.org/site/guide</a>YesAdditional CommentsIs the data acquisition clear, complete and methodologically sound?YesAdditional CommentsIs there sufficient detail in the methods and data-processing steps to allow reproduction?YesAdditional CommentsIs there sufficient data validation and statistical analyses of data quality? YesAdditional CommentsIs the validation suitable for this type of data?YesAdditional CommentsIs there sufficient information for others to reuse this dataset or integrate it with other data?YesAdditional CommentsAny Additional Overall Comments to the AuthorRecommendationAccept

---

## [Reviewer Report]

Reviewer name and names of any other individual's who aided in reviewer Paul TaconetDo you understand and agree to our policy of having open and named reviews, and having your review included with the published papers. (If no, please inform the editor that you cannot review this manuscript.)YesIs the language of sufficient quality?YesPlease add additional comments on language quality to clarify if needed
Are all data available and do they match the descriptions in the paper? NoAdditional CommentsThere are notable differences between the level of detail presented in the manuscript and that available in the dataset. While the manuscript provides detailed information on species composition, sporozoite infection rates, sources of blood meals, and so on, the dataset contains very limited data, mainly restricted to the year and location of sample collection. From what I understand (not sure though), there were 1062 sampling sessions over which 477111 anopheles mosquitoes were collected. However in the dataset ‘occurrence’ what you actually report are the ‘events’ (ie. mostly the dates and places of collections). To be compliant with the GBIF requirements (and overall to make the dataset more useful), you should use the Gbif "Sampling Event" standard (https://www.gbif.org/data-quality-requirements-sampling-events), with 2 datasets : 'event' (describing the sampling event) and 'occurrence' (describing the actual occurrences of the species, related to the sampling event). In the 'event' dataset, please report : - the exact date of collection (colum EventDate) (not only the year). - the sampling effort (colums sampleSizeValue, sampleSizeUnit, SamplingEffort) and in the 'occurrence' dataset : - the number of individuals collected for each event (column individualCount) In addition you may fill the column municipality (easier for re-use) Also, where available, you should : - report the occurrence at the species level (not only the genus, as it is the case in the current version of the dataset). - report the additional information (Sporozoites Infectivity, Blood Meal Analysis, etc..)Are the data and metadata consistent with relevant minimum information or reporting standards? See GigaDB checklists for examples <a href="http://gigadb.org/site/guide" target="_blank">http://gigadb.org/site/guide</a>YesAdditional CommentsIs the data acquisition clear, complete and methodologically sound?YesAdditional CommentsIs there sufficient detail in the methods and data-processing steps to allow reproduction?YesAdditional CommentsIs there sufficient data validation and statistical analyses of data quality? YesAdditional CommentsIs the validation suitable for this type of data?YesAdditional CommentsIs there sufficient information for others to reuse this dataset or integrate it with other data?YesAdditional Commentsin the abstract, please add the collection time frame (start date and end date)Any Additional Overall Comments to the AuthorThank you for your motivation in sharing your very valuable dataset !RecommendationMinor Revision